# Numerical Study of Liquid Metal Turbulent Heat Transfer in Cross-Flow Tube Banks

**Alessandro Tassone** [1,*] , **Jasper Meeusen** [2], **Andrea Serafini** [1] **and Gianfranco Caruso** [1]

1   Department of Astronautical, Electrical and Energy Engineering—DIAEE, Nuclear Engineering Research Group, Sapienza University of Rome, Corso Vittorio Emanuele II, 244, 00186 Roma, Italy

2   Department of Mechanical Engineering (TME), KU Leuven, Celestijnenlaan 300A–postbus 2421, B-3001 Heverlee, Belgium

*   Correspondence: alessandro.tassone@uniroma1.it

**Abstract:** Heavy liquid metals (HLM) are attractive coolants for nuclear fission and fusion applications due to their excellent thermal properties. In these reactors, a high coolant flow rate must be processed in compact heat exchangers, and as such, it may be convenient to have the HLM flowing on the shell side of a helical coil steam generator. Technical knowledge about HLM turbulent heat transfer in cross-flow tube bundles is rather limited, and this paper aims to investigate the suitability of Reynolds Average Navier–Stokes (RANS) models for the simulation of this problem. Staggered and in-line finite tube bundles are considered for compact ($a = 1.25$), medium ($a = 1.45$), and wide ($a = 1.65$) pitch ratios. The lead bismuth eutectic alloy with $Pr = 2.21 \times 10^{-2}$ is considered as the working fluid. A 2D computational domain is used relying on the $k - \omega$ Shear Stress Transport (SST) for the turbulent momentum flux and the $Pr_t$ concept for the turbulent heat flux prediction. The effect of uniform and spatially varying $Pr_t$ assumptions has been investigated. For the in-line bundle, unsteady $k - \omega$ SST/$Pr_t = 0.85$ has been found to significantly underpredict the integral heat transfer with regard to theory, featuring a good to acceptable agreement for wide bundles and $Pe \geq 1150$. For the staggered tube bank, steady $k - \omega$ SST and a spatially varying $Pr_t$ has been the best modeling strategy featuring a good to excellent agreement for medium and wide bundles. A poor agreement for compact bundles has been observed for all the models considered.

**Keywords:** heat transfer; turbulent Prandtl number; liquid metal; cross-flow; tube bank

## 1. Introduction

Liquid metals (LM) are attractive candidates for applications in nuclear fission and fusion reactors. Sodium (Na), lead (Pb), and lead–bismuth eutectic alloy (PbBi or LBE) are studied as coolants for applications in Generation-IV Fast Breeder Reactors (GEN-IV FBR) thanks to their excellent thermal properties and low moderation capability. In fusion reactors, tin (Sn), lithium (Li), and lead–lithium eutectic alloy (PbLi) are considered as armor to protect plasma-facing components (Sn and Li), coolant (all), and tritium breeding medium (only PbLi). Common drawbacks associated with liquid metal use for nuclear applications are their chemical reactivity (Na/Li), toxicity (Pb/PbLi), corrosivity (Pb/PbLi), and interaction with electromagnetic fields (all, only for fusion applications) [1,2].

There is a growing interest in the nuclear community for heat exchangers that extract the thermal power carried by the LM and convey it to a secondary fluid, being that water, helium, or carbon dioxide [3–5]. In recent years, the attention has been focused on heat exchangers where the LM is flowing shell side and the heat transfer regime can be described as an external cross-flow on tube banks. Contrariwise to the significant knowledge base accrued for heat transfer in sub-assembly tube bundles, in which the coolant is flowing parallel to the heated element axis [2,6], these configurations are poorly studied in the literature.

Due to the lack of experimental data, numerical simulations with Computational Fluid Dynamics (CFD) codes can be an important tool to support the design of these components. However, the accurate prediction of heat transfer rate for a LM turbulent flow is challenging due to the low Prandtl number (Pr $\ll$ 1) of these fluids, which makes it problematic to assume a similarity in the momentum and heat transport processes [6]. Nevertheless, most commercial CFD codes provide turbulence models that are built upon this assumption and use the concept of the turbulent Prandtl number (Pr$_t$) to calculate local turbulent heat fluxes.

Numerical simulation of turbulent heat transfer within cross-flow tube banks is important also for practical applications involving conventional fluids (Pr $\geq$ 1), for whichthe validation of numerical models is usually performed using benchmarks such as the Simonin and Barcouda [7] and the Balabani experiments [8], both performed using water, which collect both integral and local measurements of turbulent quantities. Some notable numerical studies are reported in Refs. [9–13]. It should be noted that various numerical techniques and strategies are used in these studies, ranging from relatively simple 2D Reynolds-Averaged Navier–Stokes (RANS) to computationally expensive 3D Large Eddy Simulations (LES), and that they vary remarkably in their accuracy in terms of prediction of local and integral quantities.

In this study, we aim to assess the suitability of a general-purpose CFD code, ANSYS Fluent, in the simulation of turbulent low-Pr heat transfer in cross-flow tube bundles. In particular, we are interested in investigating the code reliability in the evaluation of the integral heat transfer. One of the main motivations for this work has been to reassess the results reported by Abramov et al., who suggested that a good agreement in terms of heat transfer prediction can be achieved for an in-line confined bundle with a relatively simple 2D URANS model [14]. If confirmed, this result could be important from a component design perspective due to the moderate computational cost compared with more sophisticated modeling approaches recommended for the simulation of conventional fluid flow in tube banks. To the best of our knowledge, numerical works dealing with low-Pr fluid cross-flow in tube banks are rare. A literature search has found hardly any study for in-line bundles, beside Ref. [14], and none at all for the staggered configuration.

A 2D numerical model realized is used to recreate Abramov et al.'s results, and the study is then extended to triangular and rectangular tube bundles in a parameter range suitable to represent common values encountered for both fission and fusion applications, i.e., Pe $= 7.670 \times 10^2$–$1.350 \times 10^3$ and $S/D = 1.25$–$1.65$ [5]. Throughout the paper, the lead–bismuth eutectic alloy (LBE), characterized by Pr $= 2.21 \times 10^{-2}$, is adopted as the modeling fluid.To assess the quality of the produced numerical results for this more general case, experimental and semi-analytical correlations for a turbulent cross-flow with Pr $\ll$ 1 fluid are used as a reference [15,16]. Both Steady-RANS (S-RANS, in which mean quantities are not allowed to evolve in time) and unsteady RANS (URANS) modeling approaches are tested for uniform and spatially varying Pr$_t$ models.

## 2. Problem Statement

The turbulent flow of an incompressible and Newtonian fluid is described completely by the set of conservation equations, i.e., mass, momentum, and energy. Using the Reynolds decomposition, variables can be expressed as the sum of a mean, time-averaged, term and a fluctuating component and, considering the velocity $\boldsymbol{u} = (u, v)$ and temperature $T$, we have

$$u = U + u_1, \ v = V + u_2, \ T = T + T'. \tag{1}$$

In a 2D space, the governing equations can be written in terms of mean and fluctuating quantities

$$\frac{\partial U}{\partial x} + \frac{\partial V}{\partial y} = 0 \tag{2}$$

$$\frac{\partial U}{\partial t} + \nabla \cdot (U\boldsymbol{u}) = -\frac{1}{\rho}\frac{\partial p}{\partial x} + \nu \nabla \cdot \left(\frac{\partial U}{\partial x}\right)$$
$$+ \frac{1}{\rho}\left[-\frac{\partial(\rho u_1^2)}{\partial x} - \frac{\partial(\rho u_1 u_2)}{\partial y}\right] \tag{3a}$$

$$\frac{\partial V}{\partial t} + \nabla \cdot (V\boldsymbol{u}) = -\frac{1}{\rho}\frac{\partial p}{\partial y} + \nu \nabla \cdot \left(\frac{\partial V}{\partial y}\right)$$
$$+ \frac{1}{\rho}\left[-\frac{\partial(\rho u_1 u_2)}{\partial x} - \frac{\partial(\rho u_2^2)}{\partial y}\right] \tag{3b}$$

$$\frac{\partial T}{\partial t} + \nabla \cdot (T\boldsymbol{u}) = \nabla \cdot (\alpha \nabla T) + \left[-\frac{\partial(u_1 T')}{\partial x} - \frac{\partial(u_2 T')}{\partial x}\right] \tag{4}$$

Equations (2)–(4) are called the Reynolds-averaged Navier–Stokes equations, in which $\rho$, $\nu$, and $\alpha$ stand for density, kinematic viscosity and thermal diffusivity. Additional terms appear there that include the fluctuating quantities and are called turbulent momentum (TMF) and heat fluxes (THF). A turbulence model gives closure to the Equations (2)–(4) by providing a way to calculate them. Commercial CFD codes such as Fluent offer a great variety of turbulence models for the calculation of TMF, whereas THFs are usually calculated assuming similarity between turbulent momentum and heat transfer features, so that they are linked by a dimensionless quantity called turbulent Prandtl number, $\mathrm{Pr}_t$, which for conventional fluid with $\mathrm{Pr} \approx 1$ can be assumed to be spatially invariant [6,17].

This assumption is no longer valid for a low Pr fluid such as a liquid metal for which heat and momentum transport features hardly share any similarity. The thicker thermal boundary layer enhances the impact of the heat flux in the conductive sub-layer such that, in most nuclear applications, the fluid is in the transition zone between conductive- and convection-dominated heat regimes. The large thermal diffusivity allows the transport of significant energy with even moderate velocity which causes the dampening of temperature oscillations at small scales and the onset of peculiar buoyancy-affected regimes [6]. Accurate heat transfer predictions are challenging to make when relying on the assumption of a constant and uniform $\mathrm{Pr}_t$. Correlations that aim to recreate the spatial distribution of $\mathrm{Pr}_t$ have been proposed by several authors [18] as well as more complex turbulent heat flux models [6].

Due to the relative scarcity of previous works, it is not clear what are the minimum requirements to achieve acceptable accuracy for numerical simulation of low Pr fluid turbulent heat transfer in cross-flow bundles. The only paper published on the topic, to our best knowledge, is the one by Abramov et al. [14] in which a good agreement with experimental results is reported for a URANS model with $\mathrm{Pr}_t = 0.85$. In this paper, we have limited our investigation to a single TMF RANS model, the $k - \omega$ Shear Stress Transport (SST) by Menter [19], for both steady and unsteady approaches. This model has been selected due to its acceptable performances reported in the literature for the case of finite/confined bundles [12,13] and the lower computational cost compared with RSTM.

Regarding the THF modeling, we consider both constant and space-dependent $\mathrm{Pr}_t$. For the former, the values suggested by Cheng and Tak for sub-channel ($\mathrm{Pr}_t = 1.5$ for $\mathrm{Pe} < 2 \times 10^3$ [20]) and pipe flow ($\mathrm{Pr}_t = 4.12$ for $\mathrm{Pe} < 1 \times 10^3$ [21]) are adopted. For a spatially varying $\mathrm{Pr}_t$, we have chosen the one suggested by Kays (Equation (18) of Ref. [18]) among the many correlations available in the literature. It can be written as

$$\mathrm{Pr}_t = \frac{f}{\mathrm{Pe}_t} + 0.85 \tag{5}$$

where $\mathrm{Pe}_t = \mathrm{Pr}\,\mu_t/\mu$ stands for the turbulent Péclet number and $f = 2$. Equation (5) was originally developed from experimental data gathered for internal flow forced convection

and was found to agree well with the measured heat transfer for $4 \times 10^3 \leq \text{Re} \leq 1 \times 10^6$. Another formulation of Equation (5), which adopted $f = 0.7$, has been found to perform well when paired with the $k - \omega$ SST turbulence model [22], and this oriented our decision, even if we retained $f = 2$ due to the better agreement with experimental data compared with $f = 0.7$.

Two planar tube bank arrangements, shown in Figure 1, have been considered: in-line or rectangular, in which the lattice is defined by a transverse ($a = S_T/D$) and a longitudinal ($b = S_L/D$) pitch ratio, and staggered or triangular, in which a diagonal ($S_D/D$) pitch ratio is added to the parameter list. This last quantity can be derived by the other two characteristic dimensions, $S_D/D = [(0.5a)^2 + b^2]^{0.5}$. Our discussion is limited to equilateral bank lattices, i.e., $a = b$ for in-line and $b = \sqrt{3}/2\, a$ for staggered, with $a = 1.25,\ 1.45,\ 1.65$ for a total of six test cases. The rod diameter is fixed at $D$ = 16 mm.

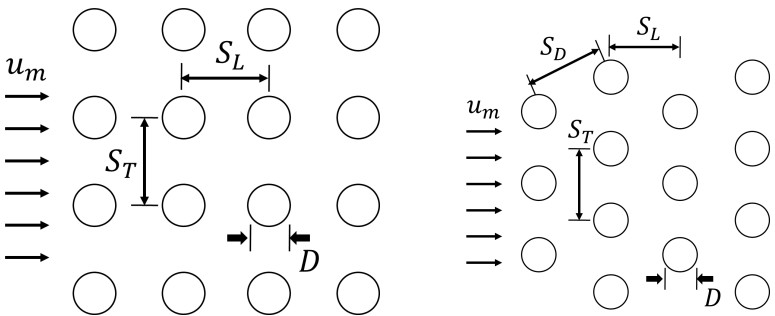

(**a**) In-line or rectangular bank      (**b**) Staggered or triangular bank

**Figure 1.** Tube bank geometry and main parameters.

The bulk interstitial velocity $u_M$, defined as the mean velocity at the minimum bank cross-section area, is used as a reference for the calculation of the dimensionless groups. This quantity is correlated to the mean superficial velocity $u_m$, that is the average fluid velocity in the absence of pipes, through the relation

$$u_M = \frac{a}{a-1} u_m. \tag{6}$$

Accordingly, we define $\text{Re} = u_M D/\nu$, $\text{Pr} = \nu/\alpha$, and $\text{Pe} = u_M D/\alpha$.

## 3. Numerical Model

The study is performed using the general purpose CFD code ANSYS Fluent. The tube bank is modeled with a 2D representation, thus assuming that the tube length is much larger than its diameter, $L \gg D$. LBE thermophysical properties, collected in Table 1, are implemented as temperature-independent and are evaluated at $T_0 = 573\,\text{K}$ from the correlations suggested in Ref. [23]. The numerical model is validated against the experimental data reported by Abramov et al. and their numerical results [14] for the case of an in-line bank with a single heated tube.

**Table 1.** LBE properties evaluated at $T_0$ from correlations in [23].

| Property | Symbol | Value | Unit |
|---|---|---|---|
| Density | $\rho$ | $1.0324 \times 10^4$ | $\text{kg m}^{-3}$ |
| Dynamic viscosity | $\mu$ | $1.8 \times 10^{-3}$ | $\text{Pa s}$ |
| Thermal conductivity | $\lambda$ | $1.1793 \times 10^1$ | $\text{W m}^{-1}\,\text{K}^{-1}$ |
| Thermal capacity | $c_p$ | $1.4494 \times 10^2$ | $\text{J kg}^{-1}\,\text{K}^{-1}$ |
| Prandtl number | $\text{Pr}$ | $2.21 \times 10^{-2}$ | - |

In 2015, Abramov et al. reported that it was possible to recreate experimental data gathered for such a case with reasonable accuracy using Fluent. The $k - \omega$ SST and $Pr_t = 0.85$ were used as TMF and THF models following an URANS approach [14]. In their study, Abramov et al. considered two tube banks: a compact layout ($a = 1.23$, $a \times b = 1.4514$, whose geometry is shown in Figure 2), and a wider configuration ($a = b = 1.69$). The numerical model was composed by a $10 \times 3$ streamwise-transverse cylinder array where only the sixth tube rank was heated by a uniform heat flux ($q_w = 1 \times 10^3$ kW m$^{-2}$). The other tubes were adiabatic, as well as the semi-circular displacers placed at the model top/bottom surface. These choices were made by Abramov et al. to develop a numerical model as close as possible to the experimental setup whose measurements they were attempting to recreate. No-slip was enforced on the pipe and top/bottom surfaces, where it is combined with symmetry for the inter-tubular spaces between half-cylinders. The fluid, LBE, entered the computational domain from the left with uniform velocity ($u_m$) and temperature ($T_0$) and exited at the right, where a pressure-outlet was imposed. The inlet boundary conditions for the turbulent quantities are chosen according to the Fluent default settings, i.e., turbulence intensity $I_u = u_i / U = 0.05$ and viscosity ratio $\mu^* = \mu_t / \mu = 10$. The mean velocity and dimensionless number range for the validation case are collected in Table 2.

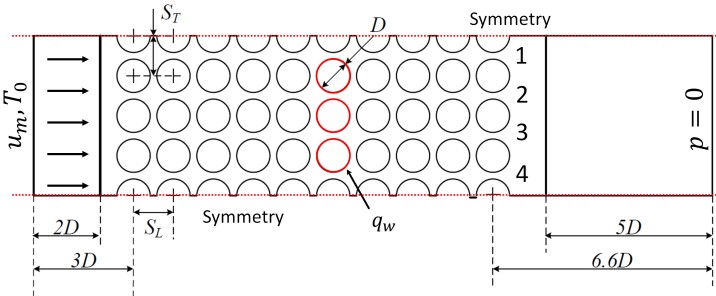

**Figure 2.** Geometry and boundary conditions for the numerical model used to recreate the results of Abramov et al. for the compact bank [14]. The vertical spaces between adjacent rods are called vanes and are labeled, in descending order, from no. 1 to no. 4.

**Table 2.** Parameter range for the validation test case.

| $u_m$ (m s$^{-1}$) | $u_M$ (m s$^{-1}$) | Re | Pe |
|---|---|---|---|
| 0.1574 | 0.8421 | $2.61 \times 10^4$ | 600 |
| 0.2099 | 1.1228 | $3.48 \times 10^4$ | 800 |
| 0.2624 | 1.4035 | $4.35 \times 10^4$ | 1000 |
| 0.3163 | 1.6915 | $5.52 \times 10^4$ | 1200 |
| 0.3619 | 1.9354 | $6.00 \times 10^4$ | 1380 |

A numerical model consistent with the one described in [14] has been used to perform the present analysis and is presented in Figure 2. A pressure-based solver is used to solve the governing equations. A second-order discretization scheme is adopted for pressure. Diffusive terms in the algebraic equations are discretized by the second-order central difference scheme, while convective terms are discretized with the third-order accurate QUICK scheme [24]. Pressure–velocity coupling is enforced through the PISO algorithm. The first-order implicit scheme is used for time discretization [24]. The URANS transient run is ended when all the monitored variables (for instance, the maximum temperature in the numerical model, $T_{max}$) have reached a statistical steady state in their averaged values and a sufficient number of periods (between five and ten) have been observed to obtain meaningful time-averaged quantities. Regardless of this global convergence criterion, a minimum transient time is always simulated. This value is defined from the bundle axial length and inlet velocity, $t_{min} = (9b + 9.6)D/u_m$, and a maximum transient time is defined as $t_{max} = 50 t_{min}$ to automatically terminate the simulation if global convergence is not

achieved. Internal convergence in each time step is reached when residuals fall below $10^{-3}$ ($10^{-6}$ for energy) or the limit of 20 iterations is exceeded.

In this study, both the S-RANS and URANS approaches are adopted to analyze the Abramov in-line bundle. To compare experimental and numerical results, the arithmetic average ($\text{Nu}_\Sigma$) of the $\text{Nu}_i$, calculated for each heated pipe, is considered, so that

$$\text{Nu}_i = \frac{q_w D}{\lambda(\overline{T_w} - T_0)} \tag{7a}$$

$$\text{Nu}_\Sigma = \sum_0^2 \text{Nu}_i . \tag{7b}$$

In Equation (7a), $\overline{T_w}$ is the surface- (and time-averaged, for URANS simulations) wall temperature. An unstructured tetrahedral grid with a prismatic inflation layer for pipe wall resolution is used. The dependence of the numerical results on spatial grid resolution and time step has been assessed taking as reference the numerical results from [14] at $\text{Pe} = 1.2 \times 10^3$. Results are collected in Tables 3 and 4. A good resolution in the interstitial space is required to achieve result independence from the grid resolution, and the time step must be chosen carefully to accurately predict the heat transfer rate. Mesh #2 has been adopted as reference for the spatial resolution in this study, whereas a uniform time scale ($\Delta t$ = 50 μs) has been selected for the time discretization, since it provides acceptable accuracy at a relatively reduced computational cost.

**Table 3.** Mesh sensitivity results for LBE tube bank cross-flow at $\text{Pe}_m = 1.2 \times 10^3$, $\text{Pr}_t = 0.85$ and $\Delta t$ = 50 (μs). Please note that $y^+$ indicates the distance of the first mesh element from the wall in wall length units.

| Mesh | [14] | 1 | 2 | 3 |
|---|---|---|---|---|
| $y^+$ | | 0.45 | 0.45 | 0.2 |
| Max bank el. size (mm) | | 0.65 | 0.55 | 0.15 |
| Elements | | $1.62 \times 10^5$ | $5.52 \times 10^5$ | $1.00 \times 10^6$ |
| $\text{Nu}_\Sigma$ | 26.88 | 27.39 | 25.93 | 25.83 |

**Table 4.** Time step sensitivity results for LBE tube bank cross-flow at $\text{Pe}_m = 1.2 \times 10^3$, $\text{Pr}_t = 0.85$ and Mesh #2 from Table 3.

| $\Delta t$ (μs) | Ref. [14] | 100 | 50 | 10 |
|---|---|---|---|---|
| $\text{Nu}_\Sigma$ | 26.88 | 25.77 | 25.93 | 26.10 |

At first, S-RANS simulations have been attempted for the compact and wide tube banks at $\text{Pe} = 6 \times 10^2$ and $1.2 \times 10^3$. The default THF model ($\text{Pr}_t$ = 0.85) has been adopted for this test. For both cases, we observed a very large deviation compared with the experimental data, as shown in Figure 3. This was expected since even second-order closure RANS models have difficulties in resolving the complex turbulent flow in an in-line tube bank [13]. It is interesting to observe that the code shows opposite behavior with regard to the integral heat transfer for these two cases. The average Nu is overpredicted for the wide bank, whereas it is underpredicted for the compact one. This could be explained by the inability of the S-RANS model to resolve the secondary transverse flow occurring in a compact tube bank. This phenomenon, described by several authors, causes enhanced fluid mixing and depends on the pitch/diameter ratio and the presence of walls enclosing the bundle [13,14]. Accurate secondary flow modeling requires higher numerical precision that what is possible to obtain from an S-RANS approach, and the result is a severe underestimate of the heat transfer. In the wide bank, the effect of secondary flow is reduced and the production of turbulent kinetic energy is enhanced, at the same time, by the wider pipe separation. As result, the convective heat flux predicted by the S-RANS model

grows beyond its actual contribution and it is responsible for the overprediction, which is consistent with the trend observed for the single cylinder case.

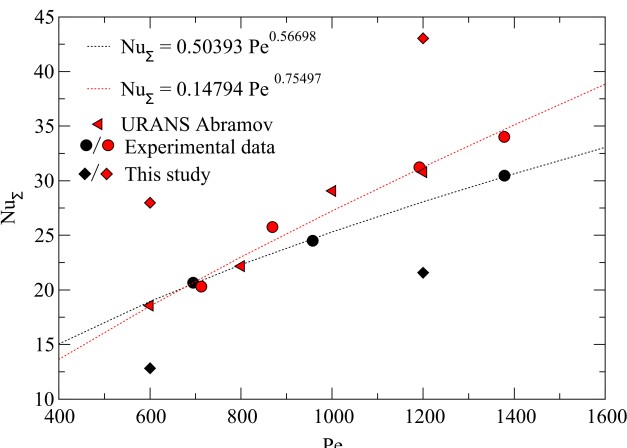

**Figure 3.** Arithmetic average $Nu_\Sigma$ plotted against Pe for S-RANS simulations: compact (black) and wide (red) tube bank. Dashed lines mark regression of experimental data presented in [14]. Relevant URANS results, also from [14], are plotted for comparison.

URANS results from [14] are represented in Figures 3 and 4 for the wide and compact bundles. An excellent agreement was described by Abramov et al. for the former case, whereas compact banks had a poor but still acceptable estimate of the average Nu for small Pe, which then improved to the same level as the wide bank when Pe was increased. It should be noted that Abramov et al. have adopted a numerical model with an increasing number of tubes in the transverse direction (1, 3, and 5) to simulate the compact bundle [14]. The results reported in Figure 4 refer to the outcome of the 3-tube model, whereas the 1-tube case underpredicted the heat transfer rate to a similar extent as our S-RANS simulation in Figure 3; no appreciable difference was observed by Abramov et al. between the 3- and 5-tube models [14]. It is unclear how many transverse tubes were present in the original test section which produced the experimental data used to validate the model, but likely no more than five.

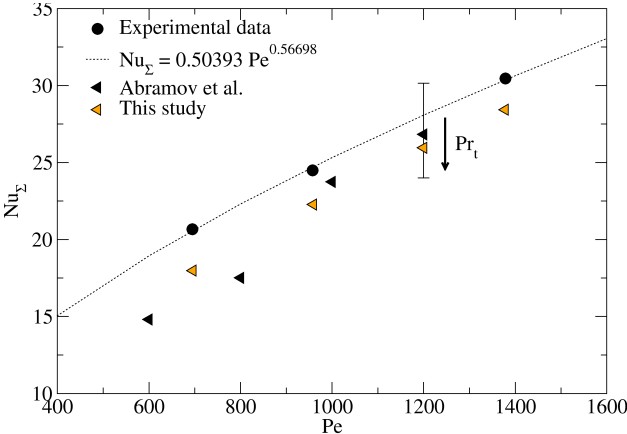

**Figure 4.** Arithmetic average $Nu_\Sigma$ plotted against Pe for URANS simulations (only compact bank). Dashed line marks regression of experimental data presented in [14]. Bar at $Pe = 1.2 \times 10^3$ is showing the result spread for $Pr_t = 0.4$–1.5.

Our URANS setup has been able to recreate the results of Abramov et al. for the compact bank, as shown in Figure 3, and even slightly outperform it for Pe = 690. The agreement with the experimental data is good with our numerical model underpredicting

the integral heat transfer by about 13% at Pe = 690 and 8% at Pe = 1380. A good agreement with Abramov et al.'s results is also found in terms of velocity field that, as shown in Figure 5, is characterized by a more significant wake swaying at small Pe, which gradually reduces due to the lowered importance of the secondary flow when Pe is increased. The characteristic behavior of the bundle wake, asymmetric at low Pe and symmetric at high Pe, is retained. That such a good agreement with experimental data could be attained without a more sophisticated THF model is surprising, but it should be stressed out that this result does not guarantee acceptable accuracy beyond just integral quantities. Abramov et al. did not provide any description of the model ability to predict local heat fluxes and temperatures compared with experimental data, probably for lack of them; therefore, no guarantees exist that these numerical results are representative of local temperature distribution in the experimental bundle from which the integral quantities were derived. Unfortunately, this will remain a limitation for the validation of numerical models until an analogue LM Simonin and Barcouda experiment is performed.

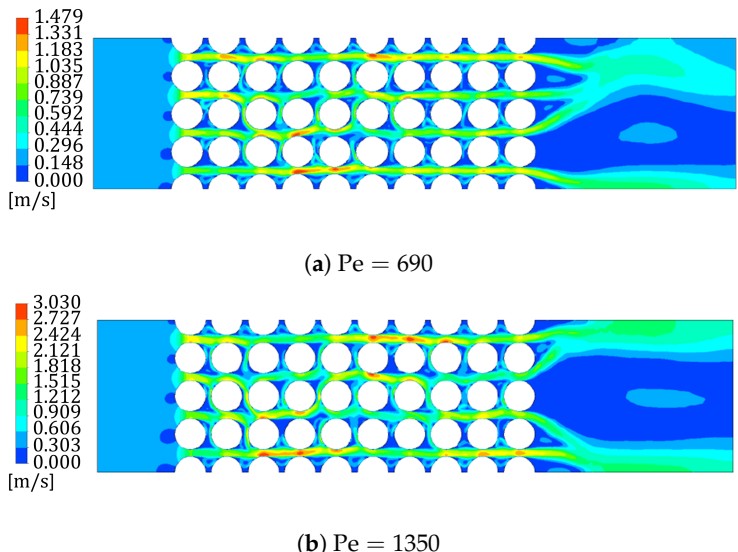

(**a**) Pe = 690

(**b**) Pe = 1350

**Figure 5.** Instantaneous velocity field from compact bank URANS simulations.

The use of a THF model relying on different assumptions for $Pr_t$ is not desirable for this case, since the already underpredicted heat transfer is made only worse when $Pr_t > 0.85$, as shown in Figure 4. Only for $Pr_t < 0.85$ the numerical results improve compared with the reference data, but this approach is not advisable since it is physically unjustified. To use a spatially varying $Pr_t$ model to improve the heat transfer prediction, a more sophisticated TMF treatment than the one considered is likely required to achieve a more accurate velocity field [6]. Nevertheless, the results produced by the present model (2D $k - \omega$ SST URANS, $Pr_t = 0.85$) have been deemed satisfactory ($\pm 15\%$ deviation from reference) for the purpose of preliminary engineering analyses and, in Section 4, it has been used to estimate the integral heat transfer in a more generic in-line tube bank, which is representative of an advanced heat exchanger.

## 4. In-Line Tube Bundle

The numerical model described in Section 3 is used to investigate three square lattice banks with $a = b = 1.25, 1.45, 1.65$, as stated in Section 2, within a range Pe = $7.67 \times 10^2$–$1.35 \times 10^3$ that, for reference, corresponds to a LBE flow with $u_M = 0.38$–$0.67$ m s$^{-1}$ and Re = $3.47 \times 10^4$–$6.11 \times 10^4$. The flow in the bundle belongs to the mixed (sub-critical) regime [25]. Boundary conditions are consistent with those shown in Figure 2 with the only difference being that the constant heat flux $q_w = 1.22 \times 10^2$ kW m$^{-2}$ is applied to all the cylinders. Semi-circular displacers are considered adiabatic. Numerical schemes, mesh, and convergence criteria are the same as those adopted in Section 3.

The analytical correlation proposed by Hsu is used to assess the numerical results [15]. For a uniform heat flux, it can be expressed as

$$\mathrm{Nu}_{th} = 0.81 \left( \frac{\phi_1}{D} \frac{u}{u_M} \mathrm{Pe} \right)^{0.5}, \tag{8}$$

where $4.2299 \geq \phi_1/D \geq 2.8569$ is the hydrodynamic potential drop for $1.25 \leq a \leq 1.65$. Tabulated values of $\phi_1/D$ can be found in Ref. [15]. The validity of this correlation has been tested by Dwyer [26] against the experimental data obtained by Subbotin et al. [27], which can be represented by a simple trend $\mathrm{Nu} = \mathrm{Pe}^{0.5}$. The two relations share the same slope, with the theoretical one underpredicting the experimental by approximately 20% over the range $10^2 \leq \mathrm{Pe} \leq 10^4$ [26]. This accuracy is deemed acceptable, particularly in light of the dependence of Equation (8) on $a$ through $\phi_1/D$. Such a phenomenon, not discernible in the data from [27], has been observed to be consistent with the general trend of the numerical data.

The time- and surface-averaged Nusselt number is evaluated for the local rod ($\mathrm{Nu}_i$) and a single column ($\mathrm{Nu}_{\Sigma,j}$) using Equation (7a,b). In Equation (7a), the bulk temperature $T_{b,j}$ is substituted to $T_0$. For the $j$-column, $T_{b,j}$ is calculated with the expression $T_{b,j} = \int_{L_j} uT dl / \int_{L_j} u dl$, where the vertical line $L_j = 4aD$ is passing through the center of the gap between the $j$ and $j-1$ (preceding) column. For the first column ($j=0$), $L_0$ is taken at $\Delta x = a/2$ upstream. A bundle-averaged Nu is defined to characterize the integral heat transfer behavior of a single test case

$$\mathrm{Nu} = \frac{1}{n-2} \sum_{j=2}^{n-1} \mathrm{Nu}_{\Sigma,j}, \tag{9}$$

where $n = 10$ stands for the number of rod columns that compose the bundle and $j = 1, 2, \ldots, n$. To partially discount the bundle entrance and exit effect, the first ($\mathrm{Nu}_{\Sigma,1}$) and last column ($\mathrm{Nu}_{\Sigma,n}$) result are discarded from Equation (9).

### 4.1. Integral Heat Transfer

Numerical results for the bundle-averaged Nu are collected in Table 5. Qualitatively, the heat transfer is increased with both Pe and $a$, consistently with the trend predicted by Equation (8). Heat transfer in in-line bundles for fluid with $\mathrm{Pr} \geq 0.7$ is usually dependent on the $a/b$ ratio [25] that, however, is constant in our case, where $a = b$. The increase in heat transfer with larger $a$ is caused by the more intense flow swaying between rows that enhances the fluid mixing, as it was already the case for Abramov et al. [14]. Deviation from the analytical relation is found to decrease with Pe, which is a behavior already observed in Section 3. For the case $a = 1.25$, our model is underpredicting the integral heat transfer from 16 to 28%, which is significantly worse than the performance recorded for the bundle in Section 3 despite a close geometric similarity ($a = 1.19$). A further deterioration of the quality of the model prediction is observed for $a = 1.45$ and $\mathrm{Pe} \leq 928$, for which the deviation reaches as much as $-45\%$. This trend is not confirmed at higher flow velocity since, for $\mathrm{Pe} > 928$, the deviation of the results from the theoretical value is consistent with the expectations. At the largest pitch ratio, $a = 1.65$, the model predicts the heat transfer with reasonable accuracy over the Pe range considered. Interestingly, for $\mathrm{Pe} \geq 1150$, the code is found to overestimate the heat transfer, which is a noticeable departure from the pattern established so far.

**Table 5.** Comparison between the bundle-averaged Nu, calculated with Equation (9), and reference values estimated with Equation (8) for the in-line bundle. Relative deviation $\epsilon = (\text{Nu} - \text{Nu}_{th}) / \text{Nu}_{th}$.

| $a$ | Pe | $\text{Nu}_{th}$ Equation (8) | Nu | $\epsilon$ (%) |
|---|---|---|---|---|
| 1.25 | 767 | 20.63 | 14.88 | −27.87 |
|  | 928 | 22.69 | 17.67 | −22.12 |
|  | 1150 | 25.26 | 21.13 | −16.35 |
|  | 1350 | 27.37 | 22.90 | −16.33 |
| 1.45 | 767 | 22.61 | 12.30 | −45.60 |
|  | 928 | 24.87 | 16.90 | −32.05 |
|  | 1150 | 27.70 | 23.63 | −14.69 |
|  | 1350 | 30.01 | 28.70 | −4.37 |
| 1.65 | 767 | 23.70 | 21.84 | −7.85 |
|  | 928 | 26.00 | 24.36 | −6.31 |
|  | 1150 | 29.03 | 30.70 | +5.75 |
|  | 1350 | 31.46 | 33.09 | +5.18 |

This behavior in the prediction of the integral heat transfer significantly differs from the one described in Ref. [14] and recreated in Section 3. This outcome, although undesirable, is not entirely surprising. A possible explanation is the different thermal boundary condition considered for this study: all rods are uniformly heated, whereas in the previous case, only column No. 6 was subjected to a uniform $q_w$. It is possible that the boundary condition considered in Ref. [14] may be particularly favorable for this kind of analysis. On the other hand, an alternative explanation is that the good performances observed are dependent on the chosen parameter space, i.e., $a$ and Pe. To support this conclusion, we may highlight that the regression of our results at $a = 1.65$ suggests a general trend $\text{Nu} \propto \text{Pe}^{0.779}$, which is consistent with the numerical and experimental data presented by Abramov et al. for their widely packed bundle ($a = 1.69$, see Figure 3) but not with Equation (8), which has a different functional dependence, i.e., $\text{Nu} \propto \text{Pe}^{0.5}$. Therefore, the relatively good results for the wide in-line bundle could be considered as mostly an effect of the Pe range investigated. A similar functional dependence, i.e., $\text{Nu} \propto Pe^{0.774}$, is observed for the numerical results of our compact bundle ($a = 1.25$), whereas the numerical and experimental trends presented by Abramov et al. are closer to Equation (8), as it is possible to see in Figure 3. The medium tube bank ($a = 1.45$) unsurprisingly deviates from both the trend of the compact and wide bundle, as it was evident by the deviation from reference values shown in Table 5, and it is characterized instead by $\text{Nu} \propto Pe^{1.5}$.

Another angle of the matter is how much we can trust Equation (8) as representative of actual physical behavior since, despite agreeing well with some experimental data, it is still a theoretical relation obtained with significant simplifications: irrotational and inviscid flow, absence of interaction between nearby thermal boundary layers, etc. The functional dependence of the integral heat transfer by Pe in a cross-flow in-line bundle has recently been assessed again by the experimental work conducted by Beznosov et al. for $a = b = 1.47$ and $900 \leq \text{Pe} \leq 2500$ [28]. Beznosov et al. found that their experimental results are well approximated by the relation

$$\text{Nu} = 5.5 + 0.025\,\text{Pe}^{0.8}. \tag{10}$$

Equation (10) features a significant departure from Equation (8) and the experimental results presented by Subbotin et al. [27], but it is closer to the behavior exhibited by our numerical model. As a final note, it is interesting to mention that our S-RANS model, previously described in Section 3 but whose results are not presented here, agrees reasonably well with Equation (10) overpredicting the integral heat transfer between 13% and 26% for $767 \leq \text{Pe} \leq 1150$. Therefore, it could be possible to have an excellent agreement with this theoretical prediction using an S-RANS model and a custom THF treatment such as the ones discussed in Section 5.

It is clear that it is difficult to draw a general conclusion from such a dispersed set of data and that a wider corpus of experimental work is necessary to assess the capability of a numerical code in predicting the heat transfer for an HLM cross-flow in-line bundle. Local temperature, velocity, and turbulence profiles are particularly desirable for this purpose but are still lacking in the literature.

### 4.2. Velocity and Temperature Distribution

The time-averaged velocity and temperature distribution for the limiting case of $a = 1.25$ and $a = 1.65$ are presented in Figures 6 and 7. The adiabatic semi-circular displacers that enclose the bundle influence the flow pattern in a way which is reminiscent of the wall-bounded configuration described by Li et. al [29]. Higher velocity is observed in vane No. 1 and No. 4, and it is placed between the displacers and the bundle proper (refer to Figure 2 for numbering), whereas the flow is comparatively suppressed in the internal vanes (No. 2 and No. 3 in Figure 2). The explanation for this behavior can be found in the flow resistance being dominated by the pressure force coefficient that, itself, is determined by the pressure distribution on the rods and, in turn, by the wake pattern [29]. The wake is swaying with time, see for example Figure 5, altering the pressure distribution on the rod and favoring fluid mixing across the vanes. In the case described by Li et al., the presence of a bounding wall suppresses this phenomenon and is responsible for the higher flow velocity in the nearby vane [29]. In our case, the displacers have a similar role with the important difference that their presence fosters a closer similarity in terms of flow pattern with an infinite bundle. For the wake of the rods close to the displacers, it is easier to sway compared with those close to a continuous wall in [29] and, as a result, the velocity overshoot in vane No. 1 and No. 4 is reduced and the bundle streamwise velocity distribution is generally more uniform, as it is demonstrated in Figure 8. No noticeable trend is observed for the velocity overshoot with regard to Pe and $a$.

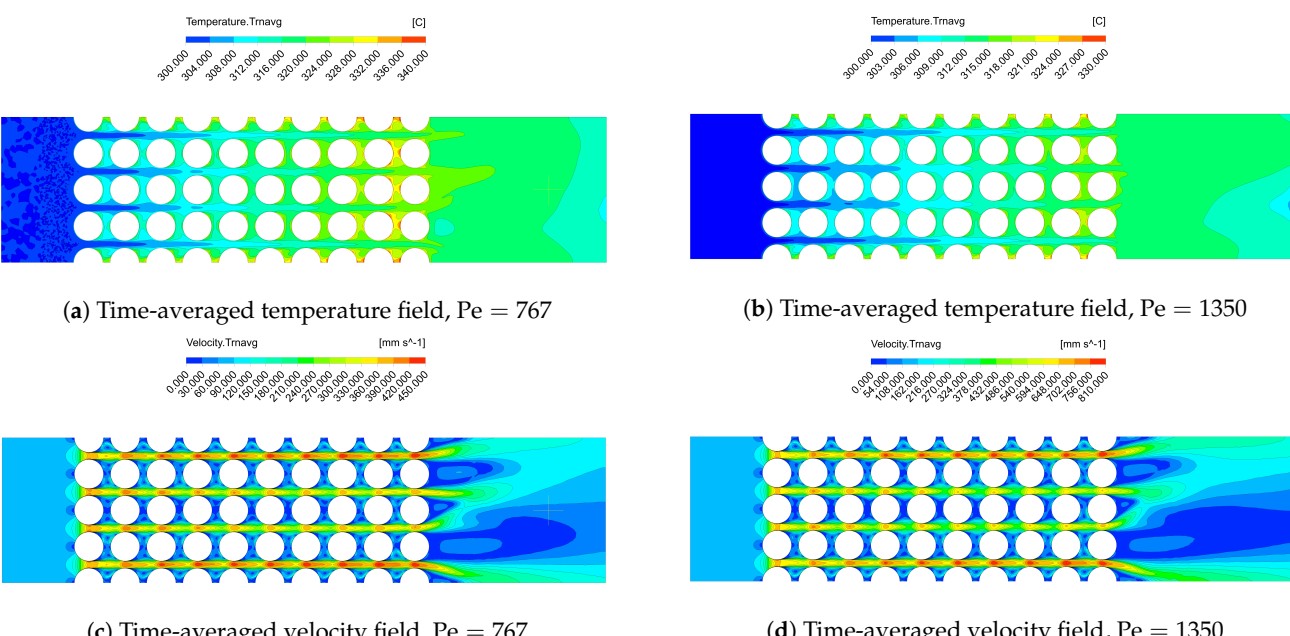

(**a**) Time-averaged temperature field, Pe = 767

(**b**) Time-averaged temperature field, Pe = 1350

(**c**) Time-averaged velocity field, Pe = 767

(**d**) Time-averaged velocity field, Pe = 1350

**Figure 6.** Numerical results for the $a = 1.25$ in-line bundle.

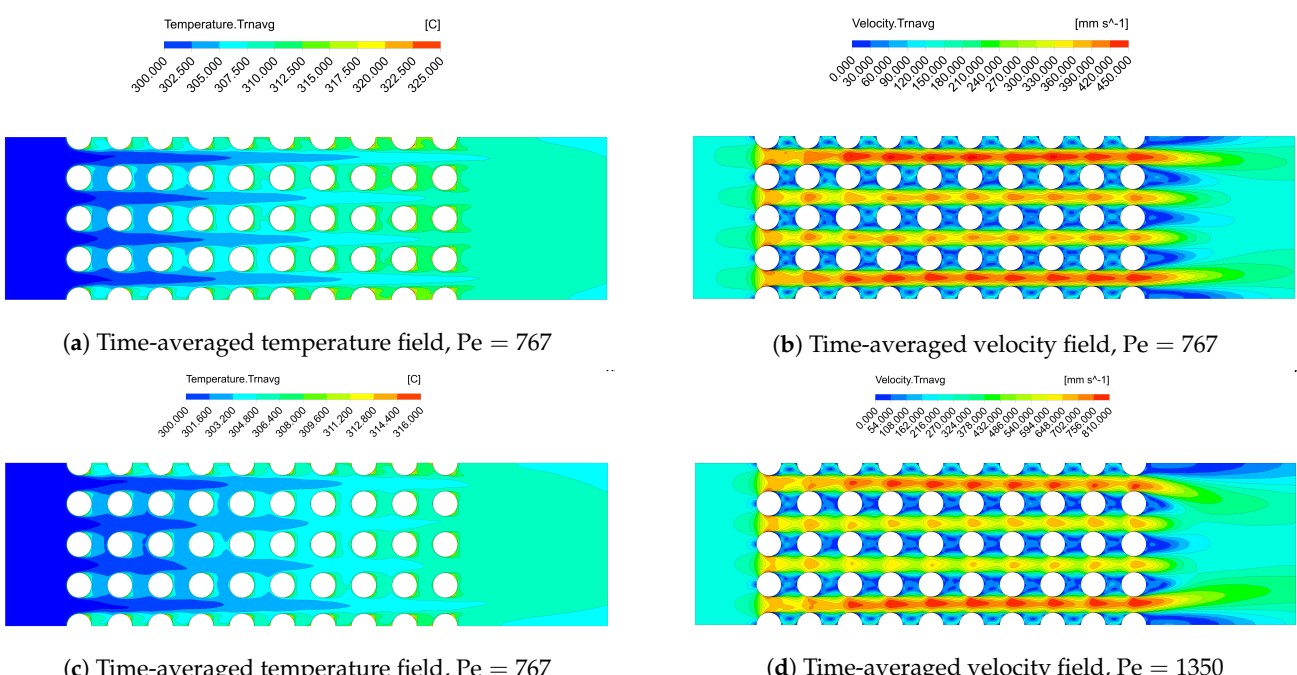

(**a**) Time-averaged temperature field, Pe = 767

(**b**) Time-averaged velocity field, Pe = 767

(**c**) Time-averaged temperature field, Pe = 767

(**d**) Time-averaged velocity field, Pe = 1350

**Figure 7.** Numerical results for the *a* = 1.65 in-line bundle.

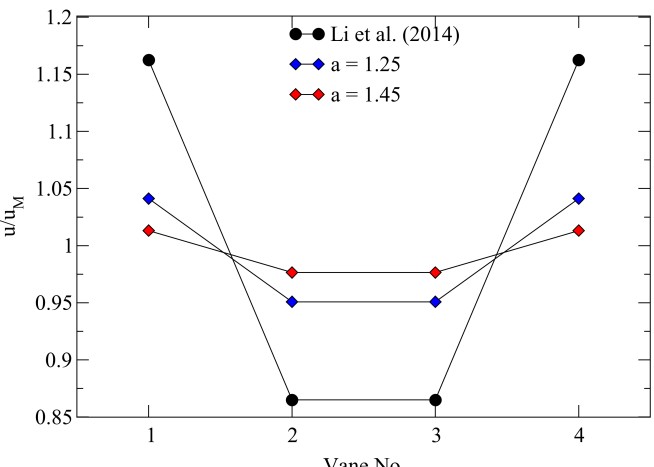

**Figure 8.** Time- and spatial-averaged velocity in the in-line bundle vanes sampled at the center of rod column No. 5 (cfr. Figure 2 for numbering) at Pe = 767 (Re = $3.47 \times 10^4$). Results are compared with data taken from Ref. [29], where Re = $3.8 \times 10^4$, *a* = 1.38 and *b* = 1.15. Vane No. 2 and 3 in the figure corresponds to vane no. 3 and 4 in Ref. [29].

The bundle wake is found to be affected by *a*. For the compact bundle *a* = 1.25, the pattern is comparable to the one observed in Ref. [14] with the presence of a large recirculation region comprised between the egress of vane No. 3 and No. 4. However, the distinct asymmetric pattern that develops at low Pe is maintained for this case even at the highest Pe investigated; see Figure 6. It is difficult to explain why no symmetrization of the wake is observed for this case since, even if this bundle is equilateral, the variation in longitudinal pitch is not significant enough to warrant this change in the flow pattern. For *a* = 1.65, the wake is found to diffuse at short distance from the bundle, see Figure 7, consistently with what was reported in Ref. [14].

Regarding the temperature distribution, Li et al. reported a general decrease in the heat transfer due to the wall effect with, correspondingly, an increase in the average temperature in the bundle vanes close to the wall [29]. In our case, the vane average temperature

distribution sampled at the rod column center is remarkably uniform for all cases, as it is also evident looking at Figures 6 and 7, which is reminiscent of the internal flow in Ref. [29]. This behavior can be explained with the fluid mixing in the external vanes not being negatively affected by the displacer presence, which is not surprising since it is their intended purpose to approximate the flow in a larger bundle.

**5. Staggered Tube Bundle**

A S-RANS $k - \omega$ SST numerical model, similar to the one described in Section 4, is used to investigate the flow and heat transfer in a staggered tube bank. The THF model consists either of a constant $Pr_t$ in the range $1.5 \leq Pr_t \leq 4.12$ or a spatially varying one using Equation (5). The bundle lattice is composed by sub-channels that are triangular in shape and equilateral, as shown in Figure 1. The pitch ratio and Pe range considered are the same as those in Section 4 as well as the thermal boundary conditions and the other settings of the numerical model. An important difference with regard to Sections 3 and 4 is that we assume the bundle to be longer in the streamwise direction, featuring a total of $n = 20$ tube columns.

A semi-empirical correlation proposed by Kalish and Dwyer is used to evaluate the prediction of the code with regard to integral heat transfer [16]. Kalish and Dwyer performed an extensive experimental study in Ref. [16] and, in particular, they investigated the heat transfer for sodium–potassium (NaK) cross-flow in staggered tube banks where all the rods are heated. The experimental data obtained have been found in good agreement with the theoretical relation proposed by Hsu [15] and were generalized for an arbitrary pitch ratio using the expression

$$\mathrm{Nu}_{th} = \left(\frac{\phi_1}{D}\right)^{0.5}\left(\frac{a^*}{a}\right)^{0.5}(5.24 - 0.225\,\mathrm{Pe}^{0.635}), \tag{11}$$

where $a^* = a - 1$ and the hydrodynamic potential drop is $3.7975 \geq \phi_1/D \geq 2.7292$ for $1.25 \leq a \leq 1.65$. Tabulated values of $\phi_1/D$ can be found in Ref. [15]. The integral heat transfer in Equation (11) is weakly dependent on $a$ and features an upward concavity. This latter condition is particularly important to achieve a better agreement with the experimental data in [16] compared with Equation (8). In particular, Kalish and Dwyer found that Equation (8) tends to underpredict heat transfer for $Pe \leq 300$ and $Pe \geq 2000$ compared with their data. This behavior was attributed to interaction between the boundary layers of the rods in the former case and significant eddy thermal transport compared with molecular conduction in the latter; both phenomena are neglected by Ref. [15]. Local average Nu at rod and column scale are evaluated consistently with the methodology described in Sections 3 and 4. The bundle averaged Nu is calculated with Equation (9) where, for this case, $n = 20$. The $T_{b,j}$ is defined for this case on the vertical line $L_j$ passing through the geometrical center of the $j$ rod column.

*5.1. Integral Heat Transfer*

The bundle-averaged Nu results for the staggered tube bank are collected in Table 6. Equation (11) shows that an increase in $a$ (and, correspondingly, $b$) is accompanied by a general increase in the heat transfer and, similarly, for Pe. For the latter, the trend of the numerical results agrees well with the theoretical relation. However, an increase in $a$ does not correspond to a rise of heat transfer in our numerical model that, instead, is usually decreasing. The only outlier moving away from this pattern is the spatially varying $Pr_t$ case. Numerical results for this THF model at $Pe < 1350$ follow the general trend, even if the spread between the lowest and highest $a$ is gradually decreasing with Pe and, finally, is inverted at $Pe = 1350$. This behavior has not been observed in Section 4.1; therefore, it is possible that this issue may be caused by the relatively coarse S-RANS model. For $a > 1.65$, the opposing trends between theoretical and numerical results are likely to cause a severe underestimation of the integral heat transfer for the $Pr_t = const$ values considered.

A spatially varying $Pr_t$ model based on Equation (5) or other relations could be more useful for these cases as well as the adoption of a more refined numerical strategy.

**Table 6.** Comparison between the bundle-averaged Nu, calculated with Equation (9), and reference values estimated with Equation (11) for the staggered bundle. Relative deviation $\epsilon = (\mathrm{Nu} - \mathrm{Nu}_{th}) / \mathrm{Nu}_{th}$.

| *a* | *Pe* | $\mathbf{Nu}_{th}$ **Equation** (11) | $\mathbf{Pr}_t$ | **Nu** | $\epsilon$ **(%)** |
|---|---|---|---|---|---|
| 1.25 | 767 | 17.88 | 1.5 | 25.64 | 43.40 |
| | | | 4.12 | 23.12 | 29.31 |
| | | | Equation (5) | 23.29 | 30.26 |
| | 928 | 19.59 | 1.5 | 27.76 | 41.70 |
| | | | 4.12 | 24.75 | 26.34 |
| | | | Equation (5) | 25.21 | 28.69 |
| | 1150 | 21.78 | 1.5 | 30.53 | 40.17 |
| | | | 4.12 | 26.84 | 23.23 |
| | | | Equation (5) | 27.85 | 27.87 |
| | 1350 | 23.63 | 1.5 | 32.81 | 38.85 |
| | | | 4.12 | 28.55 | 20.82 |
| | | | Equation (5) | 29.87 | 26.41 |
| 1.45 | 767 | 19.99 | 1.5 | 24.44 | 22.32 |
| | | | 4.12 | 21.29 | 6.56 |
| | | | Equation (5) | 22.40 | 12.11 |
| | 928 | 21.90 | 1.5 | 26.85 | 22.60 |
| | | | 4.12 | 22.96 | 4.84 |
| | | | Equation (5) | 24.55 | 12.10 |
| | 1150 | 24.36 | 1.5 | 29.73 | 22.09 |
| | | | 4.12 | 25.32 | 3.98 |
| | | | Equation (5) | 27.45 | 12.73 |
| | 1350 | 26.42 | 1.5 | 32.25 | 22.11 |
| | | | 4.12 | 26.87 | 1.74 |
| | | | Equation (5) | 29.98 | 13.52 |
| 1.65 | 767 | 21.21 | 1.5 | 23.78 | 12.12 |
| | | | 4.12 | 19.87 | −6.32 |
| | | | Equation (5) | 21.90 | 3.25 |
| | 928 | 23.24 | 1.5 | 26.26 | 12.99 |
| | | | 4.12 | 21.59 | −7.10 |
| | | | Equation (5) | 24.40 | 4.99 |
| | 1150 | 25.84 | 1.5 | 29.43 | 13.89 |
| | | | 4.12 | 23.62 | −8.59 |
| | | | Equation (5) | 27.52 | 6.50 |
| | 1350 | 28.03 | 1.5 | 31.88 | 13.74 |
| | | | 4.12 | 25.43 | −9.28 |
| | | | Equation (5) | 30.20 | 7.74 |

The quantitative agreement of the numerical heat transfer prediction with the theoretical relation is strongly dependent on *a*, Pe and the THF model adopted. The largest deviation across all models is consistently produced by $Pr_t = 1.5$. This is not entirely surprising, since this value has been recommended to model the HLM heat transfer in a fission reactor sub-channel, in which the stream direction is aligned with the heating elements [20]. For the compact bundle ($a = 1.25$), all THF models perform poorly, and the heat transfer is overestimated between $\approx 21\%$ and $43\%$. The best performances for this configuration are offered by $Pr_t = 4.12$, which is also the model most significantly affected

by Pe, with an increase generally improving its prediction. It should be noted that also the other THF models follow the same trend with Pe but to a lesser degree.

For the medium bundle ($a = 1.45$), the results for all THF models tend to better agree with the theoretical value. In particular, $\mathrm{Pr}_t = 4.12$ shows an excellent agreement with a deviation $1.5\% \leq \epsilon \leq 6.5\%$. The non-uniform $\mathrm{Pr}_t$ performs slightly worse, but it is still characterized by a good relative agreement. This last model and $\mathrm{Pr}_t = 1.5$ are pretty much insensitive to Pe, whereas $\mathrm{Pr}_t = 4.12$ confirms the trend observed for the compact bundle. Moreover, all the models are consistent in the overestimation of the heat transfer.

Results for the wide bundle ($a = 1.65$) feature a departure from this pattern, since for this case, we observe for the first time underprediction of the integral heat transfer from the $\mathrm{Pr}_t = 4.12$ model. Performance-wise, the spatially varying $\mathrm{Pr}_t$ is found to have the best agreement. However, all models show a moderate deviation, and it can be said that they are in good to excellent agreement. $\mathrm{Pr}_t = 1.5$ remains insensitive to Pe, whereas the estimate from the other two models is found to degrade if increasing it.

Overall, it can be said that only $\mathrm{Pr}_t = 4.12$ and the spatially varying $\mathrm{Pr}_t$ model offer an acceptable accuracy for the integral heat transfer estimate over the range considered. For $a = 1.25$, the heat transfer is strongly dominated by conduction due to the proximity between the tubes. The S-RANS model overestimates the contribution of the THF, and this causes the observed large deviation from the reference value. For larger $a$, the reduction of THF operated by $\mathrm{Pr}_t > 1$ is sufficient to offset the overprediction and, even for a coarse constant-value model, it is possible to obtain a very small deviation. This is particularly highlighted by the wide bundle case that suggests that even for $a \geq 1.65$, it could be possible to derive a relatively good estimate from a constant-value $\mathrm{Pr}_t$ model for an HLM heat exchanger.

The spatially varying $\mathrm{Pr}_t$ model performs surprisingly well considering that the Kays' correlation was originally developed for internal flows. The effect of pitch ratio on the $\mathrm{Pr}_t$ distribution between two nearby tubes is shown in Figure 9. Since $\mathrm{Pr}_t$ in Equation (5) depends entirely on the eddy viscosity, it reaches very high values close to the rod wall where $\mu_t$ progressively decreases and effectively $\mu_t \to 0$ in the viscous sub-layer. Conversely, $\mu_t$ reaches a finite value moving away from the wall, and so does $\mathrm{Pr}_t$, which is found to be mostly constant in the inter-space between rods. For higher Pe, we will observe a larger $\mu_t$ in this region and, progressively, the value of $\mathrm{Pr}_t$ decreases until, for very large Pe, the Reynolds analogy is recovered with $\mathrm{Pr}_t \approx 1$. The flow pattern around a rod in a staggered bundle is similar to the one observed for a single pipe [25]. For large $a$, the space between the rods is big enough, and the model performance is substantially in line with what could be observed for a single rod. Reducing $a$, this is no longer the case, and the model fails to substantially improve the integral heat transfer prediction compared with the constant $\mathrm{Pr}_t$ ones. Overall, these results seem to suggest that a spatially varying $\mathrm{Pr}_t$ could be useful to predict the integral heat transfer in a staggered bundle. Performances are quite satisfactory for $1.45 \leq \mathrm{Pr}_t \leq 1.65$, and they hint to a possible application even for $a > 1.65$. For $a < 1.45$, its use is of no benefit compared with the constant $\mathrm{Pr}_t$ model, and Equation (5) should be tweaked to account for phenomena in compact bundles.

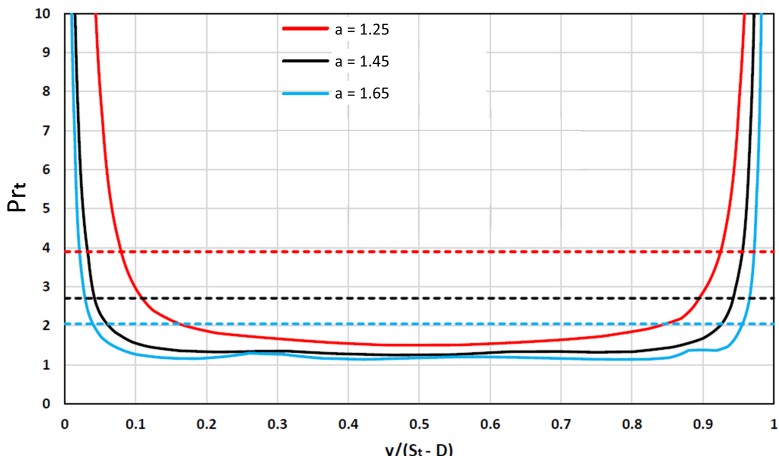

**Figure 9.** Turbulent Prandtl number distribution between two nearby pipes according to Equation (5) for Pe = 1350. Dashed lines mark the average value calculated between $0.05 \leq y/(S_T - D) \leq 0.95$.

### 5.2. Flow Pattern

The pattern in the equilateral triangle arrangement is dominated by the fluid inertia with the formation of well-defined flow lanes where the fluid smoothly moves over the rods [30]. The effect of the pitch ratio on the flow pattern is a relatively minor one and is presented in Figure 10. Counter-rotating vortexes are always formed in the rod wake, and its length tends to increase with larger $a$. For the compact bundle ($a = 1.25$), the small space available between the rods forces the wake to break down into four separated vortexes. Our results well agree with those presented by Ridluan and Tokuhiro [10]. Temperature distribution in the sub-channel centered around the central rod of the fifth column is presented in Figure 11 and is representative of the bundle pattern. Wake vortices favor a more efficient fluid mixing in the wide bundle, whereas temperatures tend to increase faster in the compact one.

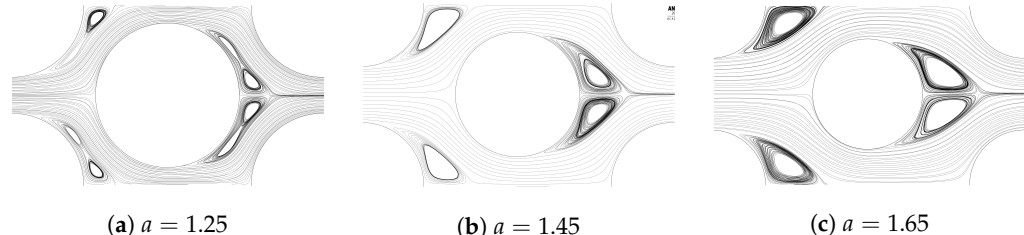

(**a**) $a = 1.25$      (**b**) $a = 1.45$      (**c**) $a = 1.65$

**Figure 10.** Velocity streamlines for the central rod of the fifth column of the staggered bundle at Re = $6.11 \times 10^4$.

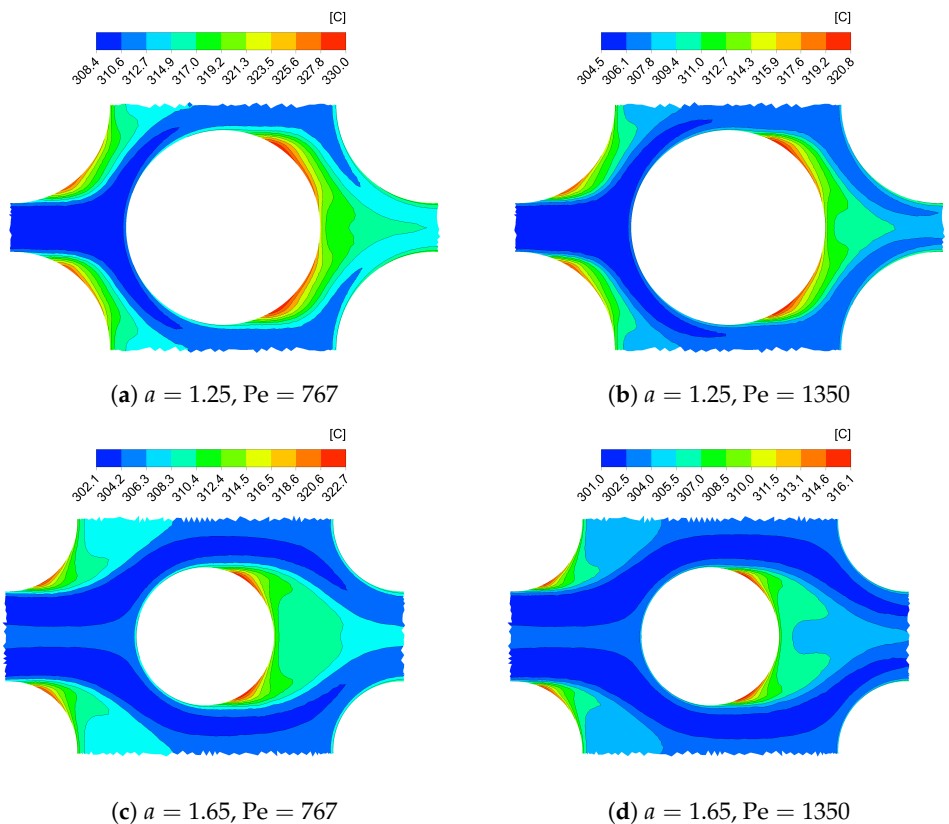

**(a)** $a = 1.25$, Pe $= 767$      **(b)** $a = 1.25$, Pe $= 1350$

**(c)** $a = 1.65$, Pe $= 767$      **(d)** $a = 1.65$, Pe $= 1350$

**Figure 11.** Temperature distribution in the subchannel centered around the central rod of the fifth column of the staggered bundle at Re $= 6.11 \times 10^4$.

## 6. Conclusions

In this study, the cross-flow of a low-Pr fluid within in-line and staggered tube bundles is investigated with the aid of the CFD code Fluent for $a$ and Pe range representative of typical configurations encountered in nuclear applications, both fission and fusion. Particularly, we have been interested in assessing the minimum computational requirements to achieve an acceptable accuracy ($\pm 20\%$) in the prediction of the integral heat transfer, which is often a very important practical concern. The analysis has been limited to a 2D model adopting S-RANS/URANS $k - \omega$ SST for the turbulent momentum flux (TMF) modeling and the $\text{Pr}_t$ concept for the turbulent heat flux (THF) modeling. The accuracy of the numerical results has been evaluated with the Hsu correlation, as shown in Equation (8), for the in-line bundle and the Kalish–Dwyer one, as shown in Equation (11), for the staggered bank. The main conclusions can be summarized as follows:

1. For a square in-line bundle, URANS calculations tend to underpredict the integral heat transfer when assuming $\text{Pr}_t = 0.85$. For a wide bundle ($a = 1.65$), the URANS approach produces a good agreement ($\pm 10\%$) with the reference value. For $a < 1.65$, the accuracy is acceptable ($\pm 20\%$) for the high end of the Pe range considered, whereas it is poor for Pe $\leq 928$. The use of uniform or spatially varying $\text{Pr}_t$ models is not advised for this case, since they will only worsen the heat transfer underprediction. S-RANS results have shown a tendency to overpredict the heat transfer for this configuration and could be revisited in the context of ad hoc $\text{Pr}_t$ treatment in a future work. For this case, the use of a more refined numerical strategy (3D URANS, LES, etc.) is recommended for an accurate estimate of the heat transfer.

2. For an equilateral triangle staggered tube bank, S-RANS calculations tend to largely overpredict the integral heat transfer at $\text{Pr}_t = 0.85$ and are amenable to treatment with different assumptions for $\text{Pr}_t$. For the compact bundle ($a = 1.25$), the ad hoc $\text{Pr}_t$ treatment still resulted in a poor agreement with the theoretical results for all the

assumptions considered. For the wide and medium tube banks, a good to excellent ($\pm 5\%$) agreement was found for a uniform $Pr_t = 4.12$ and the spatially varying Kays' model for the Pe range considered. In particular, the use of S-RANS and Equation (5) is recommended for this configuration due to its more robust behavior, even accounting for not excellent performances for compact bundles.

Although acceptable results can be obtained by the coarse TMF and THF models investigated in this study, these conclusions should be considered as only preliminary, since no attempt has been made to validate the local distribution of temperature, velocity, and turbulence quantities calculated by the numerical model in the tube bank, which will more clearly demonstrate the soundness of the heat transfer estimate. The acquisition of high quality experimental data for LM heat transfer is particularly challenging and, currently, no benchmarks exist similar to those available for other working fluids [7,8].

Future numerical work should follow along three main lines:

1. The use of more refined TMF models in conjunction with ad hoc $Pr_t$ treatment should be investigated, especially for the case of in-line bundles, for which LES is recommended even for conventional fluids.
2. The promising results obtained with the Kays' correlation for staggered bundles provide motivation to more extensively test it on a wider $a$ range and for different configurations (rotated square, equal spacing). A recently developed reformulation of the Kays' correlation, specifically intended for use with $k - \omega$ SST as the TMF model, should be evaluated as a possible alternative to the $Pr_t$ treatment [31]. More complex THF modeling, relying on additional transport equations and that have been recently been investigated in the context of sub-assembly flow [32,33], should also be explored for this configuration and the cross-flow inline tube bank.
3. It should be assessed if and how much the model performance is affected by a larger computational domain size, ideally comparing it with the condition of an infinite bundle, and how this factor affects the $Pr_t$ treatment.

**Author Contributions:** A.T.: Conceptualization, Methodology, Visualization, Writing—Original Draft. J.M.: Methodology, Investigation, Visualization. A.S.: Methodology, Investigation, Visualization. G.C.: Conceptualization, Supervision, Resources, Writing—Review and Editing. All authors have read and agreed to the published version of the manuscript.

**Funding:** This research received no external funding.

**Data Availability Statement:** Data available on reasonable request from the corresponding author.

**Acknowledgments:** The authors wish to acknowledge the contribution of Vincenzo Narcisi, who was instrumental in the definition of the test matrix used to perform this study.

**Conflicts of Interest:** The authors declare no conflict of interest.

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
