# Peer review of "Numerical Study of Liquid Metal Turbulent Heat Transfer in Cross-Flow Tube Banks"

_energies, doi:10.3390/en16010387_

Round 1

Reviewer 1 Report

1. Please show more clearly the turbulence model that is used.

2. Authors mentioned only the sixth tube was heated in page 5. Why did you do that?

3. In page 5, it is found that  2nd order discretization  was adopted for pressure and QUICK for all the other variables. Is it correct? As far as I know, QUICK is used for convective terms, not variables.

4. Please add the information for grid independence. And show the Reynolds number and boundary conditions for turbulent quantities to check the applicability of turbulence model.   

5. In page 10, the word, vane, appears in the following sentence. "Higher velocity is observed in vane No. 1 and No. 4." What this is? Please check other pages too.

6. k-w SST, SST turbulence model, SST k-w numerical model

It would be nice to organize the above terms into the correct ones. 

Author Response

We wish to thank the reviewer for the time and effort spent in reading our manuscript. Please see the attachment below for our point-by-point reply to their comments.

Reviewer 2 Report

This paper draft presents a numerical study of heat transfer in liquid metal flow in cross-flow tube banks. The method is tested on the in-line and the staggered tube arrangements and is compared to reference results from the literature. The paper draft is generally well-written and organized. I recommend the publication of the paper after the comments are considered and corrected.

1.       What is the total calculational/CPU time needed to reach the steady state in each case?

2.       Tables 4 and 5: It seems that the brackets are missing in the relative deviation formulation.

3.       Equation 9. You partially discount the effect of the bundle entrance and exit by calculating the bundle average Nu for n=10 rod columns and discounting the first and last results. Why do you divide the equation by n and not n-2?

4.       Figure 7: The captions do not match the figures.

5.       It would be nice to see the velocity and temperature distribution for the staggered grid as well. I suggest the authors add figures for time-averaged temperature and velocity fields for staggered bundles similar to those in Figures 5 or 6 for in-line bundles.

Author Response

(The authors gave the same response as above.)

Reviewer 3 Report

Summary:

In this manuscript, the authors investigate the performance of the SST turbulent model via a couple of 2-dimensional simulations of crossflow tube bundles with various arrangement patterns (in-line and triangled/staggered) and different pitch ratios. To model heat transfer characteristics, especially integral heat transfer, they employ both constant and spatial turbulent Prandtl numbers and compare the simulation results with experiments and literature. I believe, at least sometimes, the RANS model is intricate, even confusing. I would suggest using more high-fidelity model like LES. It is practicable for the 2D problems. Nevertheless, it’s an interesting study with some potentially useful industrial applications. Also, this paper is generally well-written and presents results in a clear manner. My recommendation is to accept with minor revision. I have included some comments and suggestions below in hope of helping the authors make the manuscript more complete.

1.       Page 2 line 69, what does 'S' in S-RANS stand for? Is it Steady? Please replace the first occurrence URANS with Unsteady RANS (URANS), and the first occurrence S-RANS with S###-RANS (S-RANS).

2.       Page 3, eqns 3a to 4 are not quite right. In eqn 3a, (u dot nabla) dot U is wrong, here u is the vector while U is a scalar. Should be U dU/dx + V dU/dy. Same for eqns 3b and 4. Or you may use Einstein notation to combine 3a and 3b. The advection term then becomes (U dot nabla) U.

3.       Page 3 lines 103 and 114. What is the Re in the current paper? Does it fall into the range of (4x10^3, 1x10^6)?

4.       3 Lines below line 144, what is the mean maximum temperature? Do you conduct conditional average over temperature?

5.       Table 2, y+, do you mean the first off-wall level grid height or grid resolution in the wall unit?

6.       To determine the transient time, do you need to consider the nuclear reaction time?

7.       Page 8 lines 199 & 200, is it counter-intuitive? How can we get accurate integral quantities from inaccurate local quantities? Is it because the local characteristics cannot be solved at all?

Author Response

(The authors gave the same response as above.)
